# A Composite Magnetosensitive Sorbent Based on the Expanded Graphite for the Clean-Up of Oil Spills: Synthesis and Structural Properties

Vadim M. Kadoshnikov [1], Tetyana I. Melnychenko [1], Oksana M. Arkhipenko [1], Danylo H. Tutskyi [1], Volodymyr O. Komarov [2], Leonid A. Bulavin [3] and Yuriy L. Zabulonov [1,*]

[1] State Institution "The Institute of Environmental Geochemistry of National Academy of Sciences of Ukraine", Academician Palladin Avenue, 34a, 03142 Kyiv, Ukraine; sgns_melnychenko@nas.gov.ua (T.I.M.)

[2] Central Research Institute of Armaments and Military Equipment of the Armed Forces of Ukraine, Povitroflotskyi Avenue, 28, 03049 Kyiv, Ukraine

[3] The Faculty of Physics, Taras Shevchenko National University of Kyiv, Academician Glushkov Avenue, 4, 03680 Kyiv, Ukraine

[*] Correspondence: 1952zyl@gmail.com

**Abstract:** Oil spills necessitate the development of effective methods for preventing their damaging effects on the environment. A number of physical, chemical, thermal, and biological methods are used to combat oil spills. Among them, sorption is considered to be efficient in removing thin oil films from water surfaces. Currently, there is an urgent need for simple methods of obtaining oil sorbents that include a magnetosensitive component to optimize the process of removing oil from the water surface. The purpose of the work is to obtain and research oil sorbents resistant to destruction, with increased bulk density and complex magnetosensitivity, based on thermally expanded graphite (TEG) with the inclusion of micro- and nano-particles of iron and its oxides. The structure and composition of the new composite material was characterized using scanning electron microscopy, energy dispersive X-ray analysis, X-ray diffractometry, thermogravimetric analysis, and laser diffraction particle sizing. The composite sorbent comprised TEG with the inclusion of iron-containing magnetosensitive particles. Metal-carbon nanoparticles (MCN) were used as the magnetosensitive component; they had a magnetosensitive iron core covered with a carbon shell. We used two methods of synthesis, namely (i) mechanical mixing of the TEG flakes and MCN particles, and (ii) applying a thermal shock (microwave processing) to the mixture of graphite intercalated with sulphuric acid and micro- and nanoparticles of iron and iron oxides. In the first case, MCN particles were fixed on the faces, edges, and other surface defects of the TEG flakes due to intermolecular forces, coordinate bonds, and electrostatic interaction. The strong adhesion of magnetosensitive iron/iron oxide and TEG particles in the second case was due to the mutual dissolution of iron and carbon components during the thermal shock, which formed an interfacial layer in which iron carbide is present. The presence of magnetosensitive components in the structure of the proposed oil sorbents allows the use of magnetic separation for the localization and removal of oil spills, increases the density of sorbents, and, accordingly, leads to a decrease in windage while retaining the advantageous properties of thermally expanded graphite. According to the results of laboratory studies, the efficiency of removing oil from the water surface is not lower than 95–96%.

**Keywords:** oil sorbent; thermally expanded graphite; intercalated graphite; magnetosensitive components; metal-carbon nanoparticles; micro- and nanoparticles of iron/iron oxides

## 1. Introduction

The pollution arising from crude oil and petroleum products is a serious environmental problem. Oil production and transportation is often accompanied by its spills, e.g., in water, rail, road transport, agricultural enterprises, and other industries. Especially dangerous

is the pollution of petroleum products in sea and river waters. The structure of oil spills by origin is as follows: from ships—25%, oil refineries—25%, tankers—20%, barge tanks—15%, platforms—15% [1]. To remove petroleum products from the surface of natural water bodies, physical, chemical, thermal, and biological methods are used. When spilled on the water surface, oil forms thin films, which could be effectively removed using sorbents. The oil sorbents should combine such properties as high hydrophobicity, stable positive buoyancy, minimal water absorption, low toxicity, and no negative impact on the environment. Synthetic and natural porous materials of different nature, such as ash, coke, peat, silica gel, alumina gel, natural and modified clays, foam polymers, foam glass, as well as various industrial and agricultural wastes in the form of aerogels, have been widely used [1–3].

Oil sorbents based on polymer materials are known. In [4], a method of producing a sorbent material for extracting oil from the water surface from non-woven polyethylene terephthalate (NWPET) fibers modified by attaching cross-linked polymer coatings to the surface of the fibers is considered. The NWPET fibers, which are the structural support for the applied functional coatings, are made from recycled polyethylene terephthalate bottles. Oil-absorbing coatings consisted of cross-linked homopolymers and copolymers based on octadecyl acrylate, maleic anhydride, and related esters. Cross-linked polymer meshes were synthesized by methods of both suspension and volume polymerization using divinylbenzene [4]. For the extraction of non-polar hydrocarbons, multilayer membranes made by the self-assembly method are used, which, due to the combination of superhydrophobicity and capillary action, can selectively absorb such an amount of oil that is 20 times greater than the weight of the material [5]. A method of obtaining reusable oil sorbent films made via the secondary processing of polyethylene and polypropylene waste is proposed, and their adsorption capacity is 55 g/g. SEM analysis revealed a macroporous structure with a pore size of 1–10 μm, which ensures oil sorption [6].

Mineral sorbents based on layered silicates and aluminosilicates (clays, zeolites) have a minimal impact on the environment (adsorption properties of some of them are given in Table 1 [7]), but their significant drawback is their high density, which leads to sedimentation of the spent sorbent to the bottom of the reservoir and, accordingly, to disruption of the bottom biochemical processes. At the same time, a porous ceramic based on vermiculite modified with nanocarbon was obtained, which has high hydrophobicity and, as a result, an increased (compared to other aluminosilicates) sorption capacity for oil products [7].

**Table 1.** Adsorption properties of some sorption materials based on layered silicates and aluminosilicates [7].

| Sorbent Material | Organic Liquid | Adsorption Capacity (g/g) |
| --- | --- | --- |
| Sepiolite | Motor oil | 0.174–0.184 |
| Bentonite | Motor oil | 0.150–0.176 |
| Zeolite | Motor oil | 0.166–0.192 |
| Modified diatomite | Benzene | 0.028 |

The main mechanism of sorption by layered silicates of non-polar hydrocarbons and oil in particular is their absorption by pores and capillaries located between the basal faces of adjacent crystallites. The amount of absorbed oil is determined by the volume of pores in microaggregations and, accordingly, depends on the area of the basal faces of crystallites of layered silicates [8]. Of the clay minerals, the most effective oil sorbent is kaolin, which is characterized by a highly developed basal surface of crystallites. Oil sorbents based on natural materials are environmentally friendly. However, they have low sorption efficacy, which is associated with their low hydrophobicity [1].

Sorbents based on carbon matrix have high sorption properties. Given the high hydrophobicity of carbon materials, effective fibrous carbon sorbents can be obtained by pyrolysis from plant biomass, such as common bamboo pulp fibers [9]. This sorbent has

excellent mechanical properties, ultra-low density, high hydrophobicity, and developed specific surface area. In the article [10], the results of the study of structural and sorption parameters and absorption properties of both plant raw materials and carbon sorbents obtained in the process of carbonization of wood sawdust, straw, leaves, and other plant residues at a temperature of 200–300 °C for 8–10 min are given. Carbonization leads to the improvement of structural-sorption, physico-chemical, and absorption characteristics of carbon sorbents. The sorption capacity increases by 3–5 times in comparison with the corresponding indicators of raw materials of plant origin, and carbon sorbents from sawdust of coniferous trees have 1.5–4 times higher sorption characteristics compared to sorbents obtained from straw and leaves [10].

The advantage of such sorbents is their ability to combine with magnetic components. Magnetic carbon nanofibers were obtained from nanocomposites based on bacterial cellulose. The three-dimensional (3D) interconnected structure of carbon nanofibers was evenly decorated with magnetic nanoparticles with the "nucleus-shell $Fe/Fe_3O_4$" structure [11]. The introduction of the magnetic component facilitates the removal of the spent sorbent using external magnetic fields. To strengthen the bond between the carbon matrix and the magnetic component, a rather complex method of introducing magnetite into the activated carbon matrix was proposed. Magnetic nanocomposites were synthesized using two different methods: (i) chemical co-precipitation by applying pre-synthesized $Fe_3O_4$ nanoparticles to the surface of activated carbon and (ii) by forming $Fe_3O_4$ nanoparticles in an activated carbon medium. The optimal properties were achieved for the sorbent obtained by the method of chemical co-precipitation of magnetite nanoparticles on the surface of activated carbon [12].

As a carbon matrix, thermally expanded graphite (TEG) can also be used. It is obtained by thermal shock of the intercalated graphite, which can be carried out in an electric furnace or with the help of microwave radiation [13,14]. TEG is characterised by high buoyancy and high sorption capacity to non-polar hydrocarbons, which allows it to effectively clean the water areas from oil and petroleum products. The buoyancy of TEG is due to its high porosity and hydrophobicity of the surface. Its buoyancy and the sorption efficiency are little influenced by the mineralization of water, which allows the use of TEG without the addition of excipients in marine and oceanic waters [15].

Mercury intrusion porometry was used to study the porous structure of TEG samples [13]. The results are presented in the Table 2 [13], covering a wide range from 5 to 31 m$^2$/g for the total pore area of prepared TEG samples. An increase in the exfoliation temperature from 700 to 900 °C (TEG3, TEG2, TEG1) leads to an increase in the total pore area, which is associated with an increase in the delamination of TEG with an increase in the exfoliation temperature. It was also established that an increase in the exfoliation temperature from 200 to 600 °C leads to an increase in the specific surface area of TEG from 5 to 150 m$^2$/g [13].

**Table 2.** Properties of TEG samples obtained from intercalated graphite particles of different sizes, taking into account different exfoliation temperatures [13].

| Exfoliation Temperature | 900 °C | 800 °C | 700 °C | | 900 °C | | |
|---|---|---|---|---|---|---|---|
| Graphite Intercalated Compound Size | | 35 Mesh | | 50 Mesh | 80 Mesh | 200 Mesh |
| Property | TEG1 | TEG2 | TEG3 | TEG4 | TEG5 | TEG6 |
| Bulk density, g/cm$^3$ | $0.0038 \pm 0.0002$ | $0.0044 \pm 0.0001$ | $0.0049 \pm 0.0001$ | $0.0094 \pm 0.0002$ | $0.0166 \pm 0.0002$ | $0.0552 \pm 0.0003$ |
| Equivalent pore volume, cm$^3$/g | 262 | 226 | 203 | 105 | 59 | 17 |
| Total pore area, m$^2$/g | 31 | 28 | 25 | 27 | 13 | 5 |
| Total pore volume, cm$^3$/g | 43 | 40 | 35 | 38 | 28 | 5 |

A significant disadvantage of TEG is its low bulk density and, as a result, its particles are easily blown away by wind, which creates significant obstacles when spraying it onto the water surface. Its high hydrophobicity, which creates high affinity with non-polar hydrocarbons including oil, is an advantage. However, it also causes clumping of TEG particles and prevents the adherence of the magnetosensitive components to TEG. In addition, the introduction of micro- and nanoparticles of magnetosensitive components into the carbon matrix is often a technologically complex process. The article [16] describes the joint use of chitosan molecules as precursors of the carbon framework and the use of pyrolysis treatment to create appropriate composite materials. A chitosan-derived, submicron-sized, garnet-shaped carbon framework was designed to disperse size-controlled $Fe_3O_4$ nanoparticles using a flexible one-step pyrolysis strategy. Protected by inert gas, homogeneously mixed $Fe^{3+}$ ions and chitosan molecules are transformed in situ into homogeneously dispersed $Fe_3O_4$ nanoparticles with controlled size and spherical supporting pyrocarbon domains with high mechanical strength, respectively. There is a known method of obtaining a sorbent based on thermally expanded graphite modified with a magnetic ferrite phase, which includes impregnation of intercalated graphite particles with an aqueous solution of ferric salts. The thermal expansion of impregnated oxidized graphite was carried out in a reducing atmosphere of methane. A sample of graphite weighing 0.1–0.3 g was introduced into a hermetic quartz reactor, through which a mixture of methane (3.5% by volume) and argon was passed. The quartz reactor was placed in a mine furnace preheated to 900 °C [17].

Given the complexity of known methods of incorporating magnetosensitive components into the carbon matrix, there is an urgent need for simple methods of obtaining oil sorbents that include a magnetic component to optimize the process of removing oil from the water surface.

The purpose of this work is to obtain and research complex magnetosensitive oil sorbents resistant to destruction with increased bulk density based on thermally expanded graphite with the inclusion of micro- and nano-particles of iron and its oxides.

## 2. Materials and Methods

### 2.1. Materials

In the experimental studies, we used graphite intercalated with sulphuric acid, which contains up to 15% sulfuric acid, thermally expanded graphite (both from Zavalivskiy Graphite Ltd., the Gajvoron district of Kirovograd region), and crude oil from the Nadvirna oil refinery (Ukraine).

As a magnetosensitive component, we used:

- metal-carbon nanoparticles (MCN) manufactured by Dniprospetsstal enterprise (Zaporizhzhia, Ukraine);
- a mixture of micro- and nanoparticles of metallic iron and iron oxides, which was obtained by the plasma-chemical method.

The intercalated graphite was obtained from natural graphite of the Zavalivske deposit (Zavalivskiy Graphite Ltd., Ukraine) according to the method described in [18]. Briefly, graphite was treated with concentrated sulphuric acid at $20 \pm 5$ °C for 1 h in the presence of ammonium persulphate $(NH_4)_2S_2O_8$ as an oxidiser. The intercalated graphite was washed with distilled water to pH 6 and dried at $40 \pm 5$ °C for 6 h.

Oil of Nadvirna oil refinery (Naftokhimik Prykarpattia PubJSC, Maidanska Nadvirna, Ukraine) is from the oil deposit in the Western Ukrainian region, the characteristics of which are given in [19]: low density (848.1 kg/m$^3$), low sulphur content (0.53%), water (0.11%) and mechanical impurities (only 0.008%), boiling start $-50$ °C, pour point $+9$ °C, viscosity above 70 °C 4.22 St.

MCN was obtained as an experimental sample from Dniprospetsstal enterprise (Zaporozhye, Ukraine). It was used as received.

The synthesis of micro- and nanoparticles of metallic iron and iron oxides was carried out in a specially designed plasma-chemical reactor equipped with electrodes made of

stainless steel, liquid feeders for controlling liquid flows, and distilled water (with pH adjusted to pH 8–9 with a solution of sodium hydroxide). The interelectrode space was filled with granules of iron alloy (steel S20) at the solid-to-liquid ratio ($v/v$) from (1:1) to (1:3). A pulse voltage of 0.5–1.0 kV with a frequency of 100–200 Hz was applied to the electrodes for 30 s. The resulting dispersion was drained into the tank through the outlet pipe. After washing the sediment with distilled water to pH 6–7, it was filtered through a paper filter for slow separation of dense sediments and dried at 40–50 °C.

### 2.2. Research Methods

Metal-carbon nanoparticles was studied using thermogravimetric analysis and auto-emission scanning electron microscopy with energy dispersive X-ray analysis (SEM-EDAX)–JEOL JSM-6490LV (JEOL Ltd., Tokyo, Japan).

The magnetosensitive component, which was obtained by the plasma-chemical method, and oil sorbent with its inclusion were investigated by scanning electron microscopy, SEM (JEOL JSM-6490LV (JEOL Ltd., Japan) and X-ray diffractometry (X-ray diffractometer DRON-3M and DRON-4M (JSC "Innovation Center Bourevestnik", RF) using monochromatic Cu-$K_\alpha$ radiation and Co-$K_\alpha$ radiation, according to the standard method [20]. The particle size distribution was determined using the Mastersizer 2000 particle size analyser with the HydroS liquid dispersion module (Malvern Instruments Ltd., Malvern, UK), based on laser light diffraction.

Thermally expanded graphite and the samples of magnetosensitive oil sorbents were investigated by SEM (JEOL JSM-6490LV (JEOL Ltd., Japan)).

### 2.3. Production of Oil Sorbents

2.3.1. Thermally Expanded Graphite

To produce TEG, an MS23K3614AW/BW (SAMSUNG, Kuala Lumpur, Malaysia) microwave oven with a power of 800 W was used.

A sample of graphite intercalated with sulphuric acid was placed in a ceramic container and exposed to microwave irradiation for 30 s.

2.3.2. Magnetosensitive Oil Sorbent from TEG and MCN (Sorbent No. 1)

The composite magnetosensitive oil sorbent was obtained by mixing thermally expanded graphite and metal-carbon nanoparticles in a glass container in a ratio from (2:1) to (10:1) using a UOSLab SH 3 laboratory shaker (Ukrorgsyntez Ltd., Kyiv, Ukraine).

2.3.3. Magnetosensitive Oil Sorbent from Intercalated Graphite and a Magnetosensitive Component Obtained by the Plasma-Chemical Method (Sorbent No. 2)

A sample of graphite intercalated with sulphuric acid was mixed with micro- and nanoparticles of iron and its oxides obtained by the plasma-chemical method described in Section 2.1, in a ratio from (1:1) to (1:3) using a UOSLab SH 3 laboratory shaker. After thorough mixing, the mixture was exposed to microwave irradiation for 30 s (at the power of 800 W).

### 2.4. Determination of the Oil Sorbent Absorption Capacity

To determine the oil absorption capacity of the sorbents No. 1 and No. 2, an imitation of ocean waters with a sea salt content of 35 g/dm$^3$ was prepared by dissolving a sample of 35 g of «Dr. Nice» sea salt for baths produced by AKTO Ltd. (Kyiv, Ukraine) in 1000 mL of distilled water.

Subsequently, 10 cm$^3$ of oil was applied to the surface of the aqueous solution. Then, 0.5–0.6 g of the magnetosensitive oil sorbent was sprayed evenly from the edges of the stain to the center. The surface dispersion that was formed was exposed to a magnetic field, resulting in the formation of salt-resistant magnetosensitive conglomerates saturated with oil, which were then removed from the water surface by magnetic separation (magnetic field strength 12 kA/m). The completeness of removal of the oil pellicle from the surface

was visually assessed, and the residual content of organic substances (according to the Chemical Oxygen Demand index) was determined by dichromate oxidizability [21]. In parallel, the Chemical Oxygen Demand index of the original aqueous solution and the solution with the corresponding oil contamination was determined.

### 3. Results and Discussion

*3.1. Production and Study of the Magnetosensitive Oil Sorbent Based on Thermally Expanded Graphite and Metal-Carbon Nanoparticles (Sorbent No. 1)*

To create a magnetosensitive oil sorbent, we used TEG as a carbon matrix, which was obtained by a thermal shock of graphite intercalated with sulphuric acid using microwave irradiation.

The process and patterns of TEG formation are described in detail in [22]. In the process of heat treatment, intercalated graphite is divided into lamellas due to the intensive transition of the intercalant, which was located in the interlaminar space of graphite, into the gas phase. As a result, thermally expanded graphite is formed. It is characterized by low bulk density (~5 kg/m$^3$) and high thermal conductivity (in the axial and radial directions—from 3 to 9 and from 3 to 350 Wm$^{-1}$ K$^{-1}$) and electrical conductivity (ranging from 263.127 $\times$ 10$^3$ $\Omega^{-1}$ m$^{-1}$ to 287.217 $\times$ 10$^3$ $\Omega^{-1}$ m$^{-1}$). TEG has well developed specific surface area (50–100 m$^2$/g), high oil absorption capacity (~60 g/g), and low water absorption capacity [18,23–25]. The high specific surface area of the sorbent is due to its high porosity and complex surface relief of the TEG scales (Figure 1).

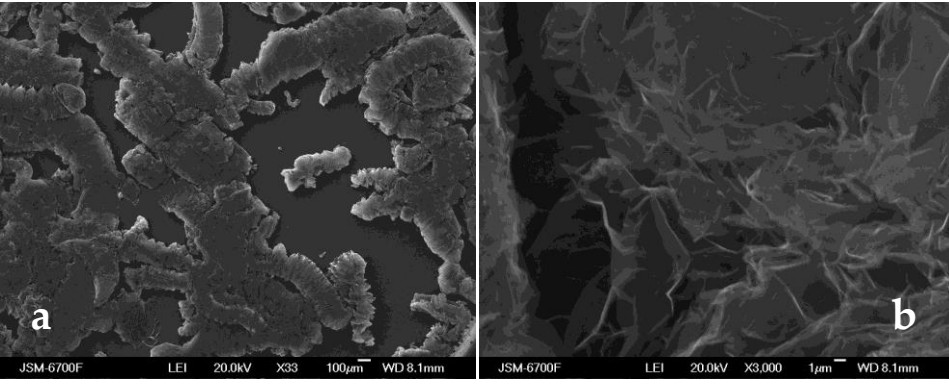

**Figure 1.** SEM micrograph of thermally expanded graphite at: (**a**) $\times$33 magnification and (**b**) $\times$3000 magnification.

According to the SEM image analysis (Figure 1a), the particle size of the formed TEG ranges from 300–400 μm to 1500–2000 μm, with the cross section from 100 to 200 μm. The outer surface of the particles (Figure 1b) is uneven and folded, and there is significant ribbing of the lateral surface of the particles, which enhances the sorption capacity of the TEG.

As a magnetosensitive component, we used MCN, which consists of a magnetosensitive core and a carbon shell. According to the results of EDAX-SEM studies (Figure 2), it was found that the magnetosensitive core consists of metallic iron doped with manganese, cobalt, nickel (total less than 2%), and iron oxide, and the size of non-aggregated particles was 1.5 μm for the metal phase and 1.4 μm for the oxidised phase. According to the results of the thermogravimetric analysis of MCN, its carbon content was 70–75%.

When mixing thermally expanded graphite and MCN, a composite sorbent was formed. It comprises a carbon matrix of TEG, on the surface of which MCN nanoparticles are firmly held (Figure 3). According to the SEM image analysis, the MCN particles are located mainly on the faces, edges, and other surface defects of the TEG particles. The strong adhesion of MCN nanoparticles to the surface of TEG flakes is due to the combination of intermolecular forces, coordinate bonds and electrostatic interaction.

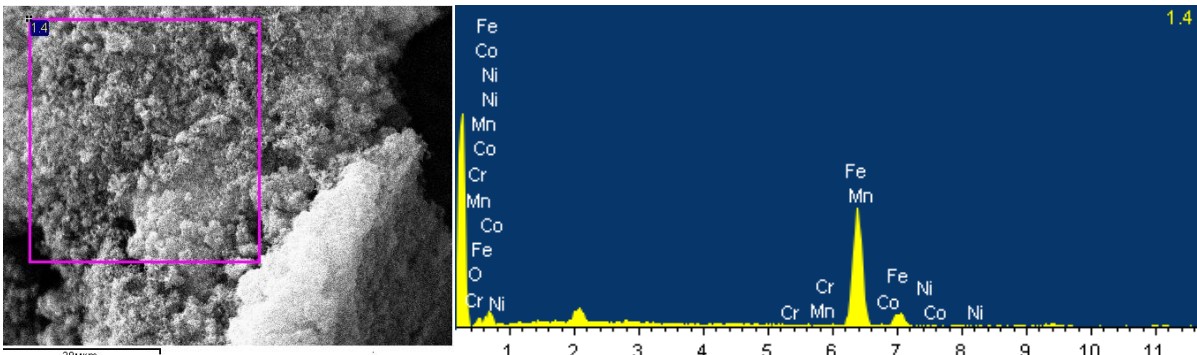

**Figure 2.** SEM-EDAX studies of the elemental composition of the metal-carbon nanoparticles.

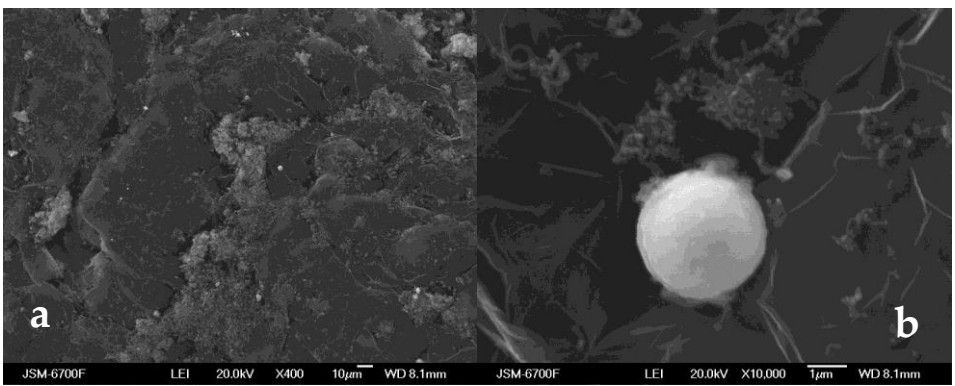

**Figure 3.** SEM micrographs of the oil sorbent No. 1 obtained by mixing of TEG and MCN at, in the ratio (2:1): (**a**) ×400 magnification and (**b**) ×10,000 magnification.

The mixing of TEG and MCN at different ratios makes it possible to obtain an oil sorbent of the required bulk density and low windage while maintaining its stable buoyancy and ensuring high sorption capacity for non-polar hydrocarbons. The main disadvantage of the resulting sorbent is its insufficient affinity between the metal core and the carbon shell within the MCN particles which could cause the loss of its integrity under the impact of individual environmental factors.

*3.2. Production and Study of Magnetosensitive Oil Sorbent from Intercalated Graphite and Magnetosensitive Component (Sorbent No. 2)*

To obtain an oil sorbent with an increased strength of the carbon-iron bond, we used the micro- and nanoparticles of iron and its oxides as a magnetosensitive component. It was synthesized by the plasma-chemical method described in Section 2.1. The granulometric composition of the resulting dispersion is shown in Table 3, and its particle size distribution is shown by Figure 4.

**Table 3.** Granulometric composition of dispersion (%), obtained by the plasma-chemical method.

| Fractions, μm | <0.1 | 0.1–1 | 1–10 | 10–100 | >100 |
|---|---|---|---|---|---|
| Sample_3, % | 3.28 | 20.78 | 51.38 | 23.70 | 0.86 |

The main granulometric fraction of the dispersion by weight is the particles from 1 to 10 μm. With regard to the number of particles, the largest fraction is in the sub-micron range, with the peak at ca. 70 nm and few particles above 500 nm (Figure 4). Taking into account that a spherical particle of 1 μm diameter is equivalent to approximately 2900 spherical particles of 70 nm in diameter in weight, it is not surprising that the count of large particles is close to zero in comparison with the number of particles in the range from 20 to 200 nm in Figure 4.

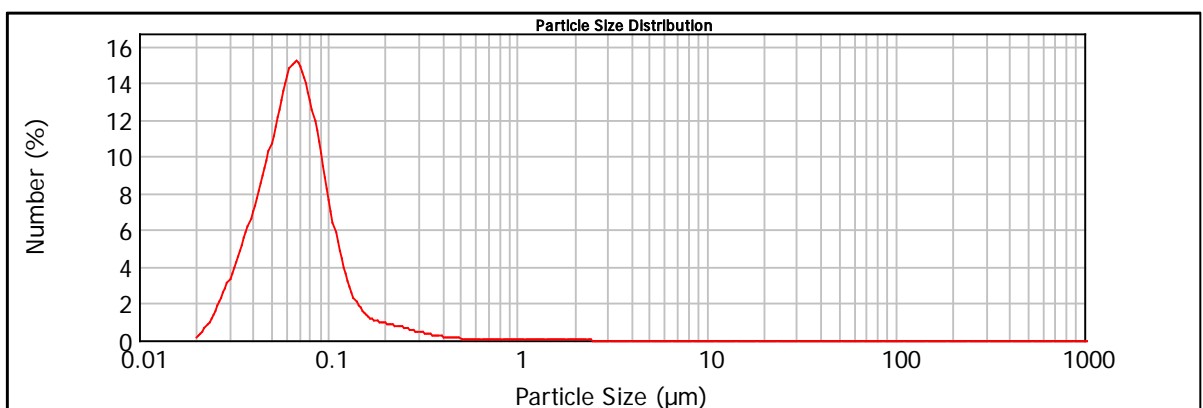

**Figure 4.** Particle size distribution of the magnetosensitive dispersion obtained by the plasma-chemical method (Sample_3).

The shape of the particles was estimated from the SEM studies (Figure 5).

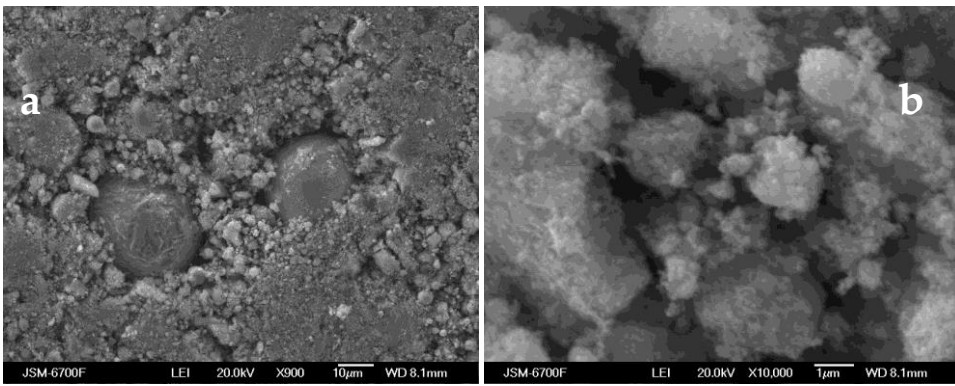

**Figure 5.** SEM micrographs of a magnetosensitive component obtained by plasma-chemical method at: (**a**) ×900 magnification and (**b**) ×10,000 magnification.

The SEM images revealed particles of irregular and spherical shape and their aggregates consisting of oxides or hydroxides of iron and ferrites in which spherical particles of iron are unevenly distributed. The metallic iron particles ensure high magnetic properties of such aggregates.

Analysis of the X-ray diffractogram (Figure 6) gives reasons to suggest that the crystalline phase of the sample is represented by weakly crystalline metal forms of $\alpha$- and $\gamma$-modifications of iron (0.2036, 0.1352 nm) and imperfect crystals of ferrihydrites $Fe_2O_3{}^*nH_2O$ (0.2036, 0.2498 nm) [26].

Iron-containing micro- and nanoparticles were mixed with intercalated sulphuric acid graphite. After thorough mixing, the mixture was exposed to microwave irradiation. As a result of the thermal shock, a magnetosensitive TEG is formed, with the iron-containing micro- and nanoparticles and their aggregates fixed on its surface (Figure 7).

In the process of heat treatment, intercalated graphite is divided into lamellas. The gases released from the intercalate interact with the particle surface, and then destroy oxides and promote the interaction of iron with carbon. During the heat treatment of the resulting mixture, partial penetration of carbon into iron is possible, and the possibility of the formation of iron carbide is also not excluded [27], which is confirmed by the diffractogram obtained by us (Figure 8a), where the reflex (0.3373 nm) belongs to graphite, and the reflex (0.1682 nm) to iron carbide [26].

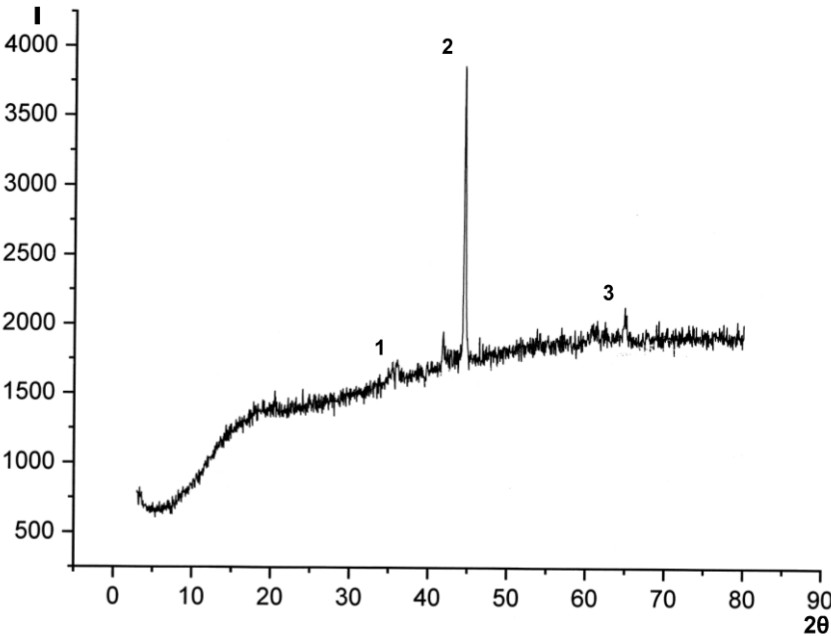

**Figure 6.** X-ray diffractogram (Cu-K$_\alpha$ radiation) of the magnetosensitive component obtained by plasma-chemical method: 1—$\alpha$-Fe$_2$O$_3$*nH$_2$O; 2—$\alpha$-Fe and $\beta$-Fe$_2$O$_3$*nH$_2$O; 3—$\gamma$-Fe.

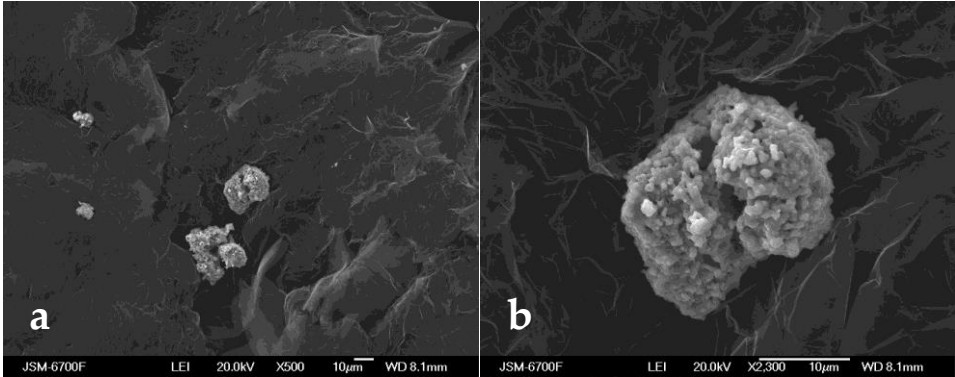

**Figure 7.** SEM images of the oil sorbent No. 2 derived from the intercalated graphite and micro- and nanoparticles of iron and iron oxides at, in the ratio (1:2): (**a**) ×500 magnification and (**b**) ×2300 magnification.

Given that a very high temperature is necessary for iron atoms to diffuse into graphite, the penetration of iron into carbon is difficult. The composition and size of the formed interphase "carbon–iron" layer depends on the specific experimental conditions [28]. As can be seen in Figure 8a, with some shift, but reflexes of $\alpha$- and $\gamma$-modifications of iron (0.2034, 0.1436 nm), ferrihydrites Fe$_2$O$_3$*nH$_2$O (0.2034, 0.2531 nm) are observed on the diffractogram of oil sorbent No. 2, and the corresponding reflexes (0.2025, 0.1432 nm and 0.2520 nm) are observed on the diffractogram of the magnetosensitive component obtained by the plasma-chemical method (Figure 8b).

The magnetosensitive TEG thus obtained is characterized by a low bulk density (~8 kg/m$^3$) and increased bonding strength of the magnetosensitive component with the carbon matrix while retaining all the advantageous properties of the TEG matrix.

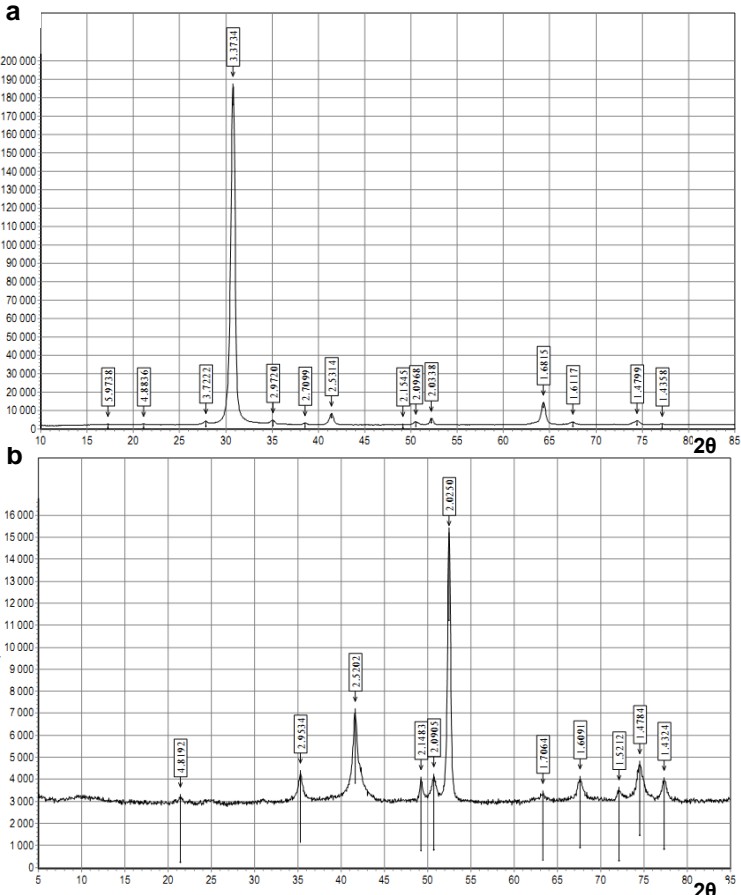

**Figure 8.** X-ray diffractogram (Co-K$_\alpha$ radiation): (**a**) of the oil sorbent No. 2 derived from the intercalated graphite and micro- and nanoparticles of iron and iron oxides at, in the ratio (1:1) and (**b**) of the magnetosensitive component obtained by plasma-chemical method.

### 3.3. Oil Absorption Capacity of the Developed Sorbents

The absorption capacity of the sorbents for oil was determined by the results of the removal of the oil film from the surface of an aqueous solution that mimics ocean water. Under the action of a magnetic field, particles of the magnetosensitive component of the sorbent are magnetized and form salt-resistant magnetosensitive conglomerates saturated with oil, which are removed from the water surface by magnetic separation. The efficiency of oil removal from the surface of the aqueous solution was assessed visually, and the residual content of organic substances in the aqueous phase was assessed by the Chemical Oxygen Demand index, the value of which for purified water was 110–140 mgO$_2$/dm$^3$. In parallel, the Chemical Oxygen Demand index of the original aqueous solution (70–80 mgO$_2$/dm$^3$) and the aqueous solution into which the same amount of oil was introduced as for the adsorption study (1700–1900 mgO$_2$/dm$^3$) was determined.

It has been experimentally established that an oil stain with an area of 1 m$^2$ (film thickness 120–150 μm) can be cleaned with 5–6 g of an oil sorbent with at least 95–96% efficiency. The process of phased removal of oil from the water surface in the laboratory using a composite magnetosensitive oil sorbent is shown in Figure 9.

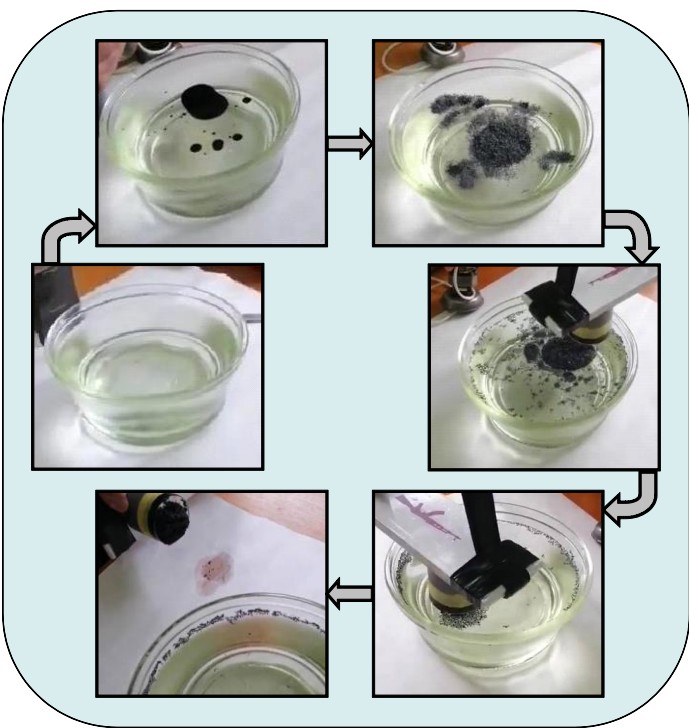

**Figure 9.** The process of removing oil from the water surface in the laboratory experiments using a composite magnetosensitive oil sorbent No. 1.

## 4. Conclusions

1. Methods to produce magnetosensitive oil sorbents, which comprise a carbon matrix (thermally expanded graphite) in combination with magnetosensitive components (metal-carbon nanoparticles or a mixture of micro- and nanoparticles of iron and its oxides), have been developed.

2. By mixing thermally expanded graphite and metal-carbon nanoparticles, a composite sorbent material is formed, which contains the TEG carbon matrix with metal-carbon nanoparticles firmly adhered to the faces, edges, and other surface defects of the graphite particles. The strong retention of metal-carbon nanoparticles on the surface of thermally expanded graphite flakes is due to the intermolecular forces, coordinate bonds, and electrostatic interactions.

3. It was found that applying the microwave irradiation to a mixture of graphite intercalated with sulphuric acid and micro- and nanoparticles of iron and its oxides, a composite sorbent with high sorption capacity for oil is formed. The strong fixation of magnetosensitive nanoparticles on the TEG carbon matrix is due to the inter-dissolution of iron and carbon during thermal shock with the formation of an interfacial layer, which helps to ensure the necessary mechanical strength of the resulting sorbent.

4. Magnetosensitive components in the structure of the proposed oil sorbents under the action of an external magnetic field form magnetosensitive agglomerates in the presence of oil, which can be efficiently removed by magnetic separation from the water surface.

5. The presence of magnetosensitive components with high density in the composite oil sorbents increases their bulk density, which can significantly reduce the dissipation of sorbent particles by wind when spraying them onto the water surface. The composite sorbents retain the advantageous sorption properties of their thermally expanded graphite matrix.

## 5. Patents

Zabulonov, Y.L., Kadoshnikov, V.M., Melnychenko, T.I., Pugach, O.V., Lytvynenko, Y.V., Shkapenko, V.V. & Odukalets, L.A. (2021). Method of removal of non-polar organic liquids from the surface of natural reservoirs and from man-made polluted waters using a complex magnetically sensitive nanosorbent. URL: https://sis.ukrpatent.org/uk/search/detail/1472524/ (accessed on: 21 February 2023).

**Author Contributions:** Conceptualization, Y.L.Z. and L.A.B.; methodology, V.M.K. and T.I.M.; formal analysis, V.O.K. and D.H.T.; data analysis, Y.L.Z. and V.M.K.; resources, Y.L.Z.; data curation, L.A.B.; writing—original draft preparation, V.M.K. and T.I.M.; writing—review and editing, V.M.K., T.I.M. and O.M.A.; supervision, Y.L.Z.; project administration, Y.L.Z. All authors have read and agreed to the published version of the manuscript.

**Funding:** The project was funded by the State Institution "The Institute of Environmental Geochemistry of National Academy of Sciences of Ukraine".

**Data Availability Statement:** Data can be provided upon request.

**Acknowledgments:** The authors express their sincere gratitude to Sergey V. Mikhalovsky, ANAMAD Ltd., Brighton, UK, for participation in the discussion of the results.

**Conflicts of Interest:** The authors declare no conflict of interest.

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
