# Peer review of "A Composite Magnetosensitive Sorbent Based on the Expanded Graphite for the Clean-Up of Oil Spills: Synthesis and Structural Properties"

_carbon, 2023_

Round 1

Reviewer 1 Report

Journal: C 

Manuscript ID: carbon-2280243

Title: A composite magnetosensitive sorbent based on the expanded graphite for the clean-up of oil spills: synthesis and structural properties

Kadoshnikov et al. aim to synthesize and study magnetosensitive sorbents based on thermally expanded graphite (TEG), which have high oil sorption capacity and stable positive buoyancy. The structure and composition of the new composite material were characterized using scanning electron microscopy, energy dispersive X-ray analysis, X-ray diffractometry, thermogravimetric analysis, and laser diffraction particle sizing. The composite sorbent comprised TEG with the inclusion of iron-containing magnetosensitive particles. Metal-carbon nanoparticles (MCN) were used as the magnetosensitive component; they had a magnetosensitive iron core covered with a carbon shell. Composite was successfully used for oil removal from water. The results are very interesting and meaningful. This topic is highly important. The manuscript fits well with the scope of the Journal. Still, it needs some improvements in order to be published. My concerns are listed below.

-The abstract lacks some exact details on results.

-The introduction is well written in general but contains very few references. A more detailed review of the literature is necessary. Also, the novelty of the proposed method should be emphasized.

-Materials and methods are ok for the characterization. On the other hand, the design of the adsorption experiment is not so clear. It should be given in more detail.

- Results of the composite characterization are well presented. Still, for the adsorption study, it is not the case. In Section 3.3. the authors should explain how did they follow the oil removal. Also, the kinetics and thermodynamics of the adsorption process would be highly beneficial to investigate.

- The conclusion is well-written and in accordance with the presented results.

Author Response

Response to Reviewer 1 Comments 

Point 1: The abstract lacks some exact details on results. 

Response 1: We took into account your comments - expanded the annotation and added additional data. 

Point 2: The introduction is well written in general but contains very few references. A more detailed review of the literature is necessary. Also, the novelty of the proposed method should be emphasized.

Response 2: Taking into account your comment, we have supplemented the introduction.

Point 3: Materials and methods are ok for the characterization. On the other hand, the design of the adsorption experiment is not so clear. It should be given in more detail.

Response 3: Changes made to section 2.4 – The completeness of removal of the oil pellicle from the surface was visually assessed, and the residual content of organic substances (according to the Chemical Oxygen Demand index) was determined by dichromate oxidizability [21]. In parallel, the Chemical Oxygen Demand index of the original aqueous solution and the solution with the corresponding oil contamination was determined.

Point 4: Results of the composite characterization are well presented. Still, for the adsorption study, it is not the case. In Section 3.3. the authors should explain how did they follow the oil removal. Also, the kinetics and thermodynamics of the adsorption process would be highly beneficial to investigate.

Response 4: Changes made to section 3.3 – The efficiency of oil removal from the surface of the aqueous solution was assessed visually, and the residual content of organic substances in the aqueous phase was assessed by the Chemical Oxygen Demand index, the value of which for purified water was 110 – 140 mgO2/dm3. In parallel, the Chemical Oxygen Demand index of the original aqueous solution (70 – 80 mgO2/dm3) and the aqueous solution into which the same amount of oil was introduced as for the adsorption study (1700 – 1900 mgO2/dm3) was determined.

The study of the kinetics and thermodynamics of the process of adsorption of oil products is not the purpose of this work, but your wish is very significant and will be taken into account in our further research.

Point 5: The conclusion is well-written and in accordance with the presented results. 

Response 5: Thank you for your appreciation of our article.

On behalf of the authors, Melnychenko T.

Reviewer 2 Report

The authors have reported the synthesis of composite sorbent based on the expanded graphite and metal-carbon nanoparticles. Some issues should be addressed before the acceptance of it.

1.      More characterizations towards sorbents No,1 and 2, including FTIR, SEM and XRD should be performed according to the different TEG and MCN contents.

2.      More experiments details about oil absorption should be performed, including the results of samples No. 1 and No. 2, the influences of dosage, intensity of the magnetic field on the oil absorption capacity should be discussed.

3.      Which kind of oil used in this paper should be pointed out.

Author Response

Response to Reviewer 2 Comments

Point 1: More characterizations towards sorbents No,1 and 2, including FTIR, SEM and XRD should be performed according to the different TEG and MCN contents. 

Response 1: Changes made to sections 3.1 and 3.2 – taking into account your comment, clarifications have been made in the captions under the SEM photo of the obtained oil sorbents – the ratio of the components (graphite and magnetosensitive component) is indicated, additionally presented is the X-ray diffractogram (Co-Кα radiation) of the oil sorbent No. 2 (figure 8).

Point 2: More experiments details about oil absorption should be performed, including the results of samples No. 1 and No. 2, the influences of dosage, intensity of the magnetic field on the oil absorption capacity should be discussed.

Response 2: Your wishes are of interest to us, but they will be taken into account in our further research

Point 3: Which kind of oil used in this paper should be pointed out.

Response 3: For research, we used oil from Nadvirna oil refinery (Naftokhimik Prykarpattia PubJSC, Ukraine) is from the oil deposit in the Western Ukrainian region, the characteristics of which are given in [19]: low density (848.1 kg/m3), low sulphur content (0.53 %), water (0.11 %) and mechanical impurities (only 0.008 %), boiling start – 50 °C, pour point +9 °C, viscosity above 70 °C 4.22 St – section 2.1.

Thank you for your careful consideration of our article.

On behalf of the authors, Melnychenko T.

Round 2

Reviewer 1 Report

The authors addressed my comments. 

Reviewer 2 Report

The manuscript can be accepted now.